# A Response Surface Methodology Approach to Improve Adhesive Bonding of Pulsed Laser Treated CFRP Composites

**DOI:** 10.3390/polym15010121

**Published:** 2022-12-28

**Authors:** Chiara Mandolfino, Lucia Cassettari, Marco Pizzorni, Stefano Saccaro, Enrico Lertora

**Affiliations:** Department of Mechanical, Energy, Management and Transport Engineering, Polytechnic School, University of Genoa, 16145 Genoa, Italy

**Keywords:** laser treatment, adhesive bonding, CFRP, design of experiments, manufacturing process optimization

## Abstract

In this work, a response surface-designed experiment approach was used to determine the optimal settings of laser treatment as a method of surface preparation for CFRP prior to bonding. A nanosecond pulsed Ytterbium-doped-fiber laser source was used in combination with a scanning system. A Face-centered Central Composite Design was used to model the tensile shear strength (TSS) of adhesive bonded joints and investigate the effects of varying three parameters, namely, power, pitch, and lateral overlap. The analysis was carried out considering different focal distances. For each set of joints, shear strength values were modeled using Response Surface Methodology (RSM) to identify the set-up parameters that gave the best performance, determining any equivalent conditions from a statistical point of view. The regression models also allow the prediction of the behavior of the joints for not experimentally tested parameter settings, within the operating domain of investigation. This aspect is particularly important in consideration of the process optimization of the manufacturing cycle since it allows the maximization of joint efficiency by limiting the energy consumption for treatment.

## 1. Introduction

### 1.1. Laser Treatment and Adhesive Bonding of CFRP

In recent years, laser technologies have become increasingly popular in a number of manufacturing fields involving the modification of material surfaces. What is attractive is the possibility of combining high values of concentrated energy with very high processing speeds, which allows surface characteristics to be tailored to specific requirements, with beam-surface interaction time that can be managed according to the material nature by selecting the appropriate laser from a wide range of sources [1,2]. As described by Issa et al. [3], it is precisely the laser-material interaction that is the crucial variable in the process, specifically the time scale of the interaction (that can be of the order of nanoseconds, picoseconds or femtoseconds) and the wavelength of the laser in relation to the characteristics of the material treated and the laser fluence. 

The preparation of substrates to produce adhesive-bonded joints is certainly one of the most promising applications of lasers. Surface preparation is correctly considered a pivotal phase to obtain good quality bonding [4,5]. In fact, after the fabrication of parts, these cannot be immediately bonded as their surfaces may be contaminated by different substances, such as dust, fingerprints, release agents, or lubricants, which would prevent the proper formation of strong and stable bonds at the adhesive-adherend interface. This fact makes prior removal of contaminants (e.g., by solvent cleaning) a prerequisite. However, a significant increase in the adhesion characteristics of surfaces can be achieved by specific processes involving the modification of morphology and roughness of the contact surfaces to better accommodate the adhesive and withstand mechanical stresses [6]. Certainly, the pre-bonding treatment must be carefully selected, based on the material being processed, the expected/required performance of the final joint, and, of course, equipment availability, and must be performed by experienced and qualified personnel.

Considering Carbon Fiber Reinforced Polymer (CFRP) composites, on which this study specifically concentrates, the most common methods for the pre-bonding treatment of adherends rely on purely mechanical actions through abrasive methods aimed at promoting mechanical interlocking, often at the expense of repeatability of results and increased contamination. This entails production aspects that cannot be underestimated: solvent cleaning is necessary both before and after treatment, which lengthens the time of a process that, being rather difficult to automate, is typically manual and, therefore, unsuitable for mass production and large surfaces. In addition, abrasive methods require more care in avoiding defects or delamination of the composite substrate, which may seriously affect joint strength. Peel ply is occasionally used as an alternative to mechanical treatments for CFRP materials. Its use, however, does not always lead to considerable increases in mechanical characteristics compared to other treatments due to a large amount of contamination and/or air trapped in the roughness profile, which causes incomplete contact and poor adhesion between adhesive and substrate [7].

In this context, laser technology has the potential to play a major role in the surface treatment of CFRP, as it offers advantages both to the treated part (contaminant removal, tailoring of surface characteristics, increased wettability) and the whole process (absence of hazardous chemicals, easy automation), and can enable improved joint performance in extremely short times and with different laser sources [8,9,10,11,12,13,14]. It should be emphasized that all these features are achieved with a completely “green” process, which is a further valuable aspect that today is increasingly pushing researchers and industry toward lasers, as it fits perfectly into the current environmental policies aimed at replacing traditional technologies (based on abrasion or chemical treatments, both for metals and polymers [15]) with more environmentally friendly and cleaner processes. As a result, the literature has also been increasingly devoted to this topic in recent years. For example, Palmieri et al. [15] compared the performance of peel-ply CFRP joints with that of joints treated with an Nd:YAG laser source. A Yb:KYW chirped pulse-regenerative amplification laser system was used by Oliveira et al. [16], who analyzed the effects of varying process parameters and carbon fiber direction on morphology. In turn, Reitz et al. [17] compared the effectiveness of ultraviolet and infrared laser pretreatments for heterogeneous CFRP-to-aluminum adhesive joints. One noteworthy aspect is that all authors agree that the use of lasers requires a thorough knowledge and experience of the process and precise optimization of the treatment parameters according to the material to be treated. Once the correct wavelength has been chosen so as not to damage the substrate, there is a huge number of laser parameters to consider in order to achieve highly controlled and repeatable surface properties: power, pulse frequency, spot size, focal distance, scanning velocity, and spot pattern, to name a few [18,19,20,21,22,23]. 

### 1.2. Background of Statistical Analysis of Adhesive Systems

To deal with the large number of process parameters inherent in laser treatment (which are also strongly dependent on the equipment used and its operating mode), the Design of Experiment (DoE) approach is certainly an indispensable tool [24], proving to be an effective support for understanding the significance of any interactions between process parameters and variables of interest [25,26,27,28]. In particular, Response Surface Methodology (RSM) can allow the behavior of the variable taken as a reference (i.e., the response) to be predicted even for untested set-ups. However, to interpret the outcomes of a statistical study aimed at optimizing the laser process for bonding, it becomes essential not to underestimate the complexity of simulating an adhesive system and to consider some necessary premises:many physical-chemical factors are involved in the adhesion phenomena at the substrate-adhesive interface, some of which are not completely controllable or predictable;many variables determine the behavior of a CFRP substrate, especially the surface conditions, which often vary depending on the matrix nature or the arrangement of reinforcing fibers, and thus are highly dependent on the manufacturing process/producer (e.g., presence of release agents, use of additives, surface finishing, etc.);since adhesive bonding in many applications is a manual process, the final quality and performance of joints are typically influenced by a human factor that cannot be reproduced or simulated.

Based on this, it is evident that to achieve the most reliable prediction of the behavior of laser-treated bonded joints, the interpretation of the data obtained from statistical analysis must be guided by a deep knowledge and understanding of the phenomena involved in the adhesion mechanisms.

To the best of the authors’ knowledge, few examples of the application of the DoE approach to the study of adhesive systems and related treatments can be found in the literature. Among these, it is worth mentioning the research of Lutey and Moroni [29], which focuses on DoE-based optimization of nanosecond pulsed laser texturing of pigmented polyethylene substrates to improve the mechanical strength of adhesive joints. In addition, in a recent study published by the same authors [30], DoE methodology was exploited to identify the effect of the parameters of low-pressure plasma surface treatment on the shear strength of adhesive bonded joints made of different substrates, including CFRP of the same type investigated here. Considering these two studies as a reference for validation of this method, in this research work, the DoE approach was applied to the study of CFRP-to-CFRP adhesive joints. A nanosecond pulsed Ytterbium Fiber Laser source was used to treat CFRP substrates to be bonded with epoxy adhesive. Two scenarios were considered, varying the focal distance of the laser beam, and for each of them, different laser parameters were considered as process variables, evaluating their influence on the tensile shear strength (TSS) of joints both experimentally and statistically. In particular, an RSM-based statistical analysis was adopted as a support to experiments to identify the optimal process settings and create a model with which to predict the joint behavior for untested treatment conditions, allowing the identification of the conditions that make the laser suitable for the treatment of CFRP substrates, combining the logics of quality, performance, and productivity.

## 2. Materials and Methods

### 2.1. Materials, Laser Setup and Joint Configuration

The CFRP used as the substrate of the adhesive bonded joints was manufactured by hand lay-up technique overlapping 7 layers of carbon fiber with a twisted 2/2 twill 0°-oriented and pre-impregnated with epoxy resin. Based on the results of preliminary tests aimed at optimizing the polymerization process, CFRP panels of a thickness of 1.50 ± 0.02 mm and Young’s modulus E equal to 70 ± 5 GPa were obtained using a vacuum bag in an autoclave for 2 h at 135 °C and pressure of 6 bar. 

Laser texturing was performed using an Ytterbium Fiber Laser source model YPLN-2–20 × 500–300 (IPG Photonics) over areas of 25 mm × 12.5 mm on the specimens by varying the average laser power (P), pitch (h), and lateral overlap (O_L_). Figure 1 provides a schematic description of the geometric parameters h and O_L_ constituting (together with power) the control factors of the statistical analysis. Their values are given as a percentage of the spot diameter (d) at the relative focal distance. The laser equipment used provides an average power output of 300 W and pulse waveforms adjustable in the range of 20–500 ns. The main characteristics and operating range of the laser setup are summarized in Table 1. 

To highlight any systematic effects of laser-parameter settings on TSS, single lap joints (SLJ) were fabricated immediately after treatment of substrates, in accordance with the geometries suggested by the ASTM D1002 [31] standard. Based on the results of preliminary research published by the authors [32,33], a commercial epoxy adhesive, DP490 produced by 3M™, was selected and applied ensuring an adhesive thickness of 0.05 mm. The sample dimensions and the bonding jig used are reported in Figure 2. Furthermore, a series of additional SLJ (from now on referred to as control joints, CJ) produced with the same bonding procedure as the laser-treated ones were realized by adopting two standard preparations: (i) solvent cleaning with acetone, and (ii) solvent cleaning followed by abrasion of the faying surfaces using a 3M™ Scotch-Brite™ MX-SR abrasive, and further acetone cleaning to remove any dust from the surface. Three specimens (n = 3) were produced per set of control joints. 

### 2.2. Surface and Mechanical Characterization

Microscopy investigation was first carried out for a preliminary observation of the phenomena and to study the effect of laser beam on the characteristics of CFRP substrates. Images of the surfaces were acquired with 50× magnification using a Leica MZ6 (Leica Microsystems, Wetzlar, Germany) optical microscope and then digitized using the dedicated X-Pro software (Ver. 8.6, Alexasoft, Florence, Italy), with which appropriate markers were added and any adjustments were made to improve image quality.

After laser texturing, all the sample surfaces were characterized with a Taylor Hobson Talyscan 150 (Taylor Hobson, Leicester, UK) non-contact automated optical profiler based on green light coherence correlation interferometry (CCI), to evaluate their topography and the influence of the various process parameters. For each parameter set-up, an area of 600 μm × 600 μm was acquired, generating a morphology map. A resolution of 340 nm on the longitudinal plane and 1 nm on the vertical axis was adopted. 

After application and curing of the adhesive, the SLJs were tested to failure in accordance with the ASTM D1002 [31] standard, by setting a test speed of 1.3 mm/min using an Instron 8802 (Instron, Norwood, MA, USA) Universal Testing machine equipped with a 50-kN load cell. TSS was calculated as the ratio between the ultimate load at failure and the initial value of overlap area. The treatment conditions were configured following the Response Surface Design approach described in the following section.

### 2.3. Statistical Evaluation of the Results Using Response Surface Methodology

A response surface design is a set of advanced Design of Experiment (DoE) techniques aimed at analyzing and optimizing a response variable, determining the optimal settings for each process factor. Central Composite Design (CCD) is the most used response surface-designed experiment. Central composite designs are factorial or fractional factorial designs with center points, augmented with a group of axial points (also called ‘star points’) to estimate curvature. A central composite design allows an efficient estimation of first- and second-order regression terms. In this study, Face-Centered Central Composite (FCC) Designs were performed. A face-centered design is a type of central composite design where the axial points are at the center of each face of the factorial space (Figure 3). This variety of design requires 3 levels for each factor. The factors investigated here were power (P), pitch (h), and lateral overlap (O_L_), varying on three levels: low and high factor levels (L1 and L2) and their midpoints (M), as reported in Table 2. Pulse frequency was fixed at 405 kHz for all tests, while Pulse duration values were optimized by the software for each combination investigated (from 20 to 120 ns). 

As a first attempt, a model of the first order (also known as Regular Two-Level) was used, characterized only by the five central points and the eight (2^3^) corner points replicated three times. However, as detailed in the following, for both factorial design models, curvature was significant, and skipping to a second-order model was necessary. An FCC design was thus performed. This approach was adopted because it allows the process results to be modeled efficiently with as few levels as possible. The selected composite design can be graphically represented using a cube with treatment combinations lying at its center, vertices, and faces, as shown in Figure 3. It was decided to run three replicates for each combination, center (C), vertices, and faces. The replication allows an estimate of the experimental error. This value is relevant to detect the actual statistical differences and distinguish them from the experimental noise affecting the collected data [34]. 

Being a very influential parameter on the amount of energy imparted to the material, the focal distance was considered as a scenario variable, and consequently, two different Response Surface Designs (RSD) were structured (Table 3): a defocused laser beam scenario referred to as Response Surface Design 1 (RSD 1), and a focused laser beam scenario referred to as Response Surface Design 2 (RSD 2). 

Analysis of Variance (ANOVA) was performed using Design-Expert 12 software (Stat-Ease, Minneapolis, MN, USA) to establish whether the effect of the process factors and their interactions had a significant impact on the response variable, which was, in this case, the TSS of adhesive-bonded joints. Precisely, the objective of this analysis was to identify individual and combined effects of the process factors on TSS. An interaction can be defined as the possible synergy of the investigated factors that occurs when the effect of a factor on the response varies, depending on the levels assumed by the other factor. Once the significant factors had been identified, TSS was modeled using Response Surface Methodology (RSM) and it was possible to determine second-order regression models representative of the response behavior in the domains selected. The response surfaces obtained then made it possible to identify the set-up parameters that maximized the TSS, considering any equivalent conditions from a statistical standpoint. The regression models also allowed us to predict the joint behavior for not experimentally tested parameter settings.

## 3. Results and Discussion

Table 4 summarizes the different process set-ups experimented with, with the relative sample designations. The same designation was used for all tests, both morphological and mechanical. Each sample number corresponds to three repetitions performed with the same parameters. 

### 3.1. Microscopy and Morphological Analyses

Prior to evaluating the effects of laser parameters, visual inspections of surfaces were carried out. In fact, heating due to the interaction between the laser beam and substrate might affect the surface negatively, entailing undesired damage such as excessive matrix burning and/or fiber failures. The pulse energy (*E*) for each parameter set-up depends on the pulse power and frequency, so both RSDs have the same energy values for each set-up. Varying the focal distance, the spot diameter (*d*) changes, resulting in the change of the fluence (*F*) parameter. The latter corresponds to the energy dose (expressed in J/cm^2^) contributed to the surface with each pulse, calculated as per Equation (1): (1)$$F=\frac{4E}{\pi d^{2}}$$

Based on the set-up parameters selected, six values of fluence (reported in Table 5) were used. Accordingly, the substrates can be divided into six macro groups in relation to the energy dose per pulse. The remarkable difference between the surfaces of samples treated using the minimum (a), and the maximum (b), fluence values is emphasized in Figure 4. When the beam hits the surface, this is not absorbed directly by the resin (which is transparent to the laser at this wavelength), but rather is absorbed by the reinforcing carbon fibers, which heat up and transfer the heat to the matrix, causing it to be removed to a greater or lesser extent depending on the beam fluence. In particular, for lower fluence values, slight traces of the beam-substrate interaction were visible, localized mainly in the portions of the surface with smaller matrix thickness; on the contrary, the adoption of higher F values caused fiber exposure in the areas above, as well as local epoxy resin burning even at the interweave zones of the reinforcement fabric, which are typical sites of matrix accumulation and thus characterized by larger amounts of resin. 

Preliminary morphological analyses confirmed a stronger interaction between the laser beam and material as the fluence increased. Table 6 depicts the morphologies of samples belonging to the said six macro groups, sorted according to F values increasing from left to right and from top to bottom. For the sake of space, the images reported are limited to one sample per fluence value, taken as representative of the relative macro-group, within which all samples exhibited nearly the same morphology after laser treatment.

Increasing the value of F, the burning of the resin caused the surface profile to locally raise at the interweave zones. With the RSD 2 samples, this behavior was certainly emphasized because the beam was focused on the substrate surface. Considering the application in adhesive bonding, this defect should be minimized since it could make it difficult to both properly couple the two faying surfaces and ensure constancy of the adhesive thickness over the bonding area. 

### 3.2. Mechanical Characterization of the Control Joints

The results of the mechanical tests conducted on the control joints (CJ) are summarized in Table 7, in which the mean value of TSS is reported together with the standard deviation (expressed in percentage terms). These outcomes were taken as a reference to compare with what was obtained after laser treatment.

### 3.3. Statistical Analysis

The average values of the TSS response variable are reported in Table 8 for both scenarios. A substantial difference in the results is noted between them. The design carried out with a defocused laser beam (RSD 1) showed TSS values higher than that of the degreased CJ in 12 out of 15 cases and higher than that of the abraded CJ with six parameter setups. In contrast, the focused-beam design (RSD 2) was characterized by the significantly worse performance of joints, showing TSS that was always lower than that of the abraded CJ and only once higher than that of the degreased-only reference. In good agreement with the morphological observations reported in Section 3.1, the mechanical behavior observed in the two scenarios reflected the fact that the excessive heating of the material, which caused local combustion and degradation of the resin that negatively affected the surface state, weakened the adhesion interactions at the substrate-adhesive interface and, consequently, lowered the shear strength of joints. However, it should be noted that different values of TSS were obtained from samples belonging to the same macrogroup, suggesting that analysis of the fluence parameter alone cannot provide complete information, as it does not take into account the influence of overlapping laser beams in the longitudinal and transverse directions [22]. Therefore, a further thorough investigation was conducted via the statistical method to determine the effect of process parameters such as pitch (h), lateral overlap (O_L_), and power (P) on the mechanical response (TSS).

#### 3.3.1. Response Surface Design 1: Defocused Scenario

The ANOVA for the defocused scenario (Table 9) resulted in a significant model. With a Face-centered Central Composite (FCC) Design, it is possible to model both first-order and second-order regression coefficients and any interactions. The FCC is a 2^k^ full factorial, to which central and 2·k points placed on the center of the sides of each square of Figure 2 have been added. An outlier value had to be eliminated to run the model, but this was not problematic, as three replicates were used for all points. 

Based on the assumptions made, a control factor—or a combination of control factors—is statistically significant if the *p*-value is less than the significance level α, chosen in this case equal to 0.05. Consequently, there were three terms producing a significant effect on the modification of TSS values, namely P, O_L_ and P^2^, while all interaction terms result in a not significant effect. In particular, the relevant parameters are listed in Table 10 in descending order of the Sum of Squares, which indicates the proportion of total variability attributable to the factor.

When a squared factor is statistically significant, its response does not change linearly, passing from the low to the high level, but its variation is quadratic, as also highlighted by following Response Surface Methodology. 

Moreover, the Lack of Fit was not significant, so the second-order linear regression model described by the related Coded Equation (Equation (2)) fitted the experimental values.
(2)$$\mathrm{TSS}=24.08-1.38\;P+3.27{\;O}_{L}+1.35{\;P}^{2}$$

This equation is representative of the expected behavior of TSS on joints fabricated with laser-treated substrates with different setups (and defocused beam). The parameters are reported in coded variables using +1 for a high level and −1 for a low level. Coding has two advantages: (i) scaling the factors to the same magnitude makes it easier to evaluate each factor’s relative importance, and (ii) it places the model intercept, β_0_, at the center of the experimental design. The signs of regression coefficients in Equation (2) indicate that increasing power (P) has a negative effect on TSS. The positive sign of the quadratic term related to power indicates the concavity of the curve, then confirmed by the relative response surfaces. In contrast, of the two geometric parameters, only lateral overlap (O_L_) was significant, and the positive sign associated with this term suggests its positive effect on TSS. The physical explanation for this can be found in the fact that defocusing the laser beam results in a reduction of the energy dose incident on the surface. The lower the energy concentration, the lower the heating of the surface and, therefore, the more limited the amount of material degraded and removed. As evidenced by morphological analysis, this situation allows the creation of a lighter but also better-defined surface texture. In this condition, the lateral overlap parameter becomes significant since, by definition, it determines the transverse pattern of the surface morphological profile, which is therefore orthogonal to the direction of application of the load acting on the joint. This causes an increase in the effective contact area between the surface and adhesive applied on it (with a consequent increase in the number of active sites for physical and chemical bonds), as well as emphasizing the phenomena of micro-mechanical adhesion at the interface, promoting mechanical interlocking. As a result, the TSS of joints increases as power decreases and O_L_ increases, the latter also being a determinant of surface local heating. 

As said, Response Surface Methodology was employed to determine the optimal settings for each factor and predict the behavior of joints in other setup conditions not tested experimentally. Figure 5 compares the response surfaces obtained by varying power and pitch and fixing the lateral overlap. 

All three surfaces model a response variable with a slight curvature. As a confirmation of the observations above, the highest TSS value was detected on the surface with O_L_ = 150%. The worst performance was found for joints #6 and #7 (see Table 8 for their TSS values), whose substrates were in fact laser treated by setting O_L_ = 50% and P = 50 W. Considering only the ANOVA analysis, the power parameter is certainly particularly significant, but likewise, spots that are greatly overlapping in the direction parallel to the laser beam result in excessive heating of the material, leading to a reduction in TSS. Consequently, both power and lateral overlap are significant factors taken individually since no combination of parameters is influential. This implies that the effects of the individual factors are independent of the level assumed by the other parameters. 

With a similar approach, it is possible to visualize the surfaces as a function of power and lateral overlap, keeping the pitch parameter constant (Figure 6). The trends of the three plotted surfaces are very similar to each other since pitch is not a significant parameter for this scenario. In turn, the analysis confirmed the positive effect of the O_L_ factor, which caused the TSS value to increase from 27.3 MPa to 30.0 MPa when combined with a low power level. 

Finally, Figure 7 displays the response surfaces plotted at constant power by varying the lateral overlap and pitch parameters. The appearance of these surfaces is similar to that of Figure 6, although no curvature is detected. However, it is evident that the maximum is found on the surface with a lateral overlap of 150% and a low power level, which is again consistent with the observations made earlier. 

In summary, from both the experimental tests and statistical model, the optimum of the domain corresponded to the setup with P = 20 W, O_L_ = 150%. There was instead a discrepancy for the pitch value: in fact, considering the predicted values, the optimal setup should be at the high level (h = 150%), while according to the experimental data, this corresponded to h at the low level (50%). However, the difference between the two maxima is not surprising because it depends on the fact that the parameter h is not significant in this model. Indeed, although the experimental values are slightly different from each other, from a statistical point of view, the ANOVA conducted shows that the recorded point variations are only due to pure experimental error (i.e., background noise), and not attributable to the variation in h (Table 11).

#### 3.3.2. Response Surface Design 2: Focused Scenario

Like the previous case, the ANOVA for the focused scenario (Table 12) resulted in a significant model. To allow comparison with RSD 1 (Table 10), the relevant parameters of RSD 2 are listed in Table 13 in descending order of the Sum of Squares. This second design was more complex compared to the previous case. Indeed, power resulted in the most significant parameter, as it was particularly influential on the value of fluence, but there were many other parameters influencing the variations in TSS, some of these appearing with quadratic effects or in combination with each other. Consequently, we can assume that power and both geometric parameters influenced the variation of the response variable.

It should be noted that all the Sum of Squares of the significant factors are lower than the Pure Error, thus indicating the presence of considerable noise and a much more perturbed and less robust experimental scenario than RSD 1. In contrast, the model accuracy of RSD 2 is higher, as deduced from the tolerance intervals of the two Response Surface Designs reported in Table 14. 

Some more information can be collected from the Coded Equation of the second model (Equation (3)):(3)$$\mathrm{TSS}\;=19.97-0.6981\;P+0.3740\;h+1.08{\;O}_{L}-0.7673{\;P}^{2}+0.9574{\;h}^{2}+0.2242{\;\mathrm{PO}}_{L}+1.18{\;P}^{2}O_{L}$$

As in the previous case, power adversely affected TSS, which decreased as the level of said factor increased. In contrast, both geometrical parameters were positive in sign, suggesting that increasing the distance between spot centers in both longitudinal and transverse directions should make TSS increase. In practice, this confirms that to ensure higher joint strength, it is necessary to limit the value of energy transferred to the substrate and heat it as little as possible. In fact, from the combined effect of the positive-sign parameters, it can be noted that power and distance between spots should be at the same level: where fluence is higher, increasing the distance between spots is needed to limit the heat transferred to the part, while the fluence reduction allows achievement of a clean, sufficiently rough surface, ensuring the micromechanical adhesion of the adhesive to the substrate. 

Certainly, the statistical analysis conducted on the individual parameters of the laser process allowed for better identification of their respective influence on the behavior of joints, providing for pivotal information to be added to that obtained by considering the effect of the fluence parameter on the surface morphology of substrates. Compared to RSD 1, the response surfaces representative of the RSD 2 scenario exhibited typical curvatures because many of the significant parameters were at the quadratic level or interaction terms. In Figure 8, the response surfaces obtained by varying power and pitch and fixing the lateral overlap are compared. Interestingly, for progressively higher values of the lateral overlap O_L_ parameter, the curve changes its concavity. Specifically, at the lowest level of O_L_ (Figure 8a), surface concavity faces downward, and for each value of h, TSS has its maximum for a mean level of power P. The surface tends to flatten when O_L_ = 100% (Figure 8b) and then assumes an upward concavity at the high level of O_L_ (Figure 8c), showing the minimum at the center of the curve (P = 35 W, h = 100%). 

This behavior is reflected by the response surfaces of Figure 9, plotted at a fixed value of pitch and at varying values of power P and lateral overlap O_L_: in each case, at low O_L_, TSS varies with power according to a downward inflected curve, whereas an upward inflected curve is described by the TSS parameter as a function of power for the higher value of O_L_. Finally, the response surfaces plotted at fixed power show similar trends to each other when varying the geometrical parameters simultaneously (Figure 10), confirming that, when setting a focused laser beam, overlap between adjacent spots in both directions must be avoided to prevent further surface damage due to excessive heating. 

### 3.4. Potential and Limitations of the Statistical Analysis of Laser-Treated Joints

The effectiveness of laser treatment against simple degreasing or mechanical abrasion is summarized in Table 15, which shows the increment/decrement percentage values of tensile shear strength (TSS), obtained by comparing, for each scenario, the expected maximum value of the domain (Maximum point—MP) determined from the fitted first-order regression model to the mean value of degreased and abraded control joints (CJ) obtained experimentally. The predictive mean of the maximum point is representative of the best behavior of a joint subjected to laser treatment within the domain investigated.

The results obtained highlighted that the efficacy of pulsed laser treatment on CFRP materials strongly depends on the focusing setup adopted for the beam. Concentrated laser spots (RSD 2), indeed, proved to be detrimental to the substrate integrity, determining excessive heating and local burning of the epoxy matrix, with consequent poor adhesion conditions at the interface with adhesive. The statistical findings evidenced that a reduction in power input along with distancing of spots in both longitudinal and transverse directions can limit the effects of the high-intensity dose of energy of the focused laser beam but are not sufficient to provide for a performance increase with respect to conventional methods. With an optimized focused laser process, in fact, the shear strength achieved by joints is comparable to that of degreased-only joints, and even lower (−7.6%) than joints simply abraded. In contrast, laser treatment can be considered a promising prebonding preparation method for CFRP materials when a defocused beam is used (RSD 1). With this setup, optimization of the process parameters can lead to joint strength up to 19.5% higher than abraded joints, achieving strength values that are comparable to those obtained with optimized plasma treatments, discussed in previous work [30].

Certainly, the possibility of obtaining improved mechanical performance of joints adopting lasers has serious implications for the economy of the entire manufacturing process, also involving environmental, technological, and production aspects that, in principle, should make lasers notably preferable to other pre-bonding methods, as introduced in Section 1. However, as discussed in this research work, obtaining appropriate treatment conditions in relation to the specific substrate to be bonded is not straightforward since the effects of different process parameters are often combined. In this context, certainly statistical analysis proved to be an excellent support to experimental tests to estimate the behavior of treated surfaces, allowing the identification of the significance of process parameters and their interactions, and suggesting possible mechanical behavior of joints outside the experimentally tested process conditions. In fact, it was precisely the statistical analysis of data that revealed that in the selected domain—especially in the optimal defocused situation—not all parameters of laser treatment studied significantly affected the TSS of joints.

On the other hand, there is no doubt that the interpretation of statistical results cannot be separated from experimental observations and a basic knowledge of the phenomena involved in the bonding process, which is, as previously introduced, seriously affected by many physical-chemical variables and manufacturing conditions that make the final behavior of joints difficult to predict. Therefore, since the aim of the treatment is to obtain the maximum TSS increase from a given adhesive system, the results of statistical analysis (which is built on experimental data) are not to be intended as absolute but are specific to the adhesive system and equipment used for surface treatment.

Having awareness of these limitations, statistics can be successfully exploited to simplify the study of complex systems such as the one investigated here, becoming a reliable method for reducing the number of tests needed, predicting joint behavior, or highlighting any hidden behavior in a certain set-up domain with an appropriate confidence interval, and hence driving production to better process conditions.

## 4. Conclusions

Ytterbium Fiber Laser treatment was performed to prepare CFRP adherends of homogeneous joints bonded with epoxy adhesive. Two scenarios (RSD 1 and RSD 2) were considered by varying the focal distance (−4 mm and 0 mm, respectively) of the laser beam. In each scenario, different laser parameters, such as fluence (F), power (P), lateral overlap (O_L_), and pitch (h) between adjacent spots, were considered as process variables, and their influence on the tensile shear strength (TSS) of joints was evaluated both experimentally and statistically. Specifically, Response Surface Methodology was adopted to identify optimal process settings and create a model to predict joint behavior for untested treatment conditions. Based on the results, the following conclusions can be drawn:A defocused laser beam (RSD 1) is preferred because the energy dose and heating of the workpiece are lower, causing limited, but more localized removal of resin from the substrate surface. Passing the pulsed laser beam in the transverse and longitudinal directions of sample surfaces results in a structured morphology. Among the geometric parameters analyzed, the O_L_ parameter was the most significant as it was responsible for the creation of surface profile traces developed in the direction orthogonal to mechanical stress, which emphasized mechanical interlocking at the adhesive-substrate interface. In contrast, no changes in joint performance were observed due to changes in the h parameter.In contrast to RSD 1, a focused laser beam (RSD 2) appeared not to be appropriate to treat CFRP composites. From the statistical analysis, power and both geometric parameters, O_L_ and h, significantly influenced TSS, since all played a role in surface heating and matrix degradation. In particular, power was the most significant parameter, negatively affecting TSS.The model developed allowed the prediction of values that had not been tested experimentally and provided the optimum condition for the two scenarios. It was found that the optimum for each scenario is obtained for the same parameter configuration, keeping power at a low level and geometric parameter(s) at a high level, i.e., P = 20 W, O_L_ = 150% (and h = 150%, for RSD 2 case). However, while an optimized defocused laser treatment can lead to a remarkable increase in joint performance (+19.5% compared to reference abraded joints), the same is not the case by adopting a focused beam setup: in fact, even at a low power level, the energy dose is so high that the performance loss due to power can only be minimized (and not completely counteracted) by increasing O_L_ and/or h parameters.Furthermore, an analysis based on surface and morphological investigation concluded that the fluence parameter should be considered since it also influences the heating of the matrix. In particular, the lower the F value, the higher the joint performance. This confirmed that optimal laser conditions are those that minimize the energy dose contributed to the CFRP substrate.

## Figures and Tables

**Figure 1 polymers-15-00121-f001:**
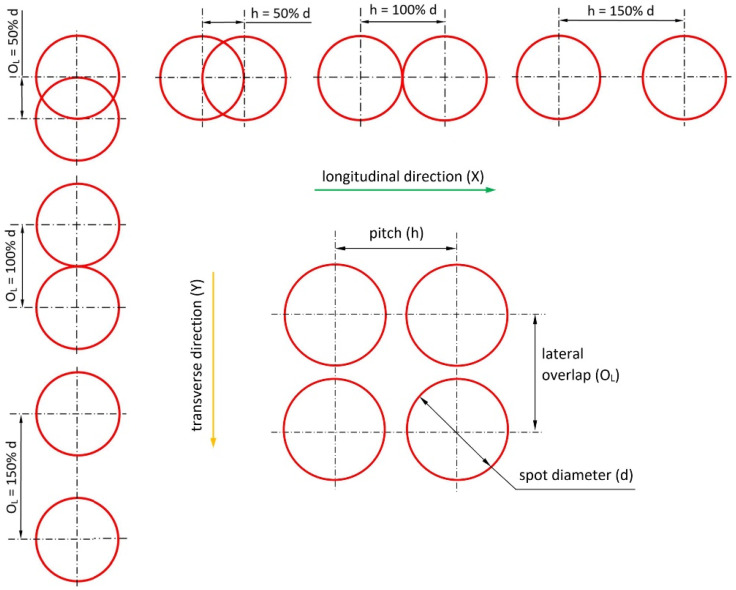
Schematic description of the geometric control factor.

**Figure 2 polymers-15-00121-f002:**
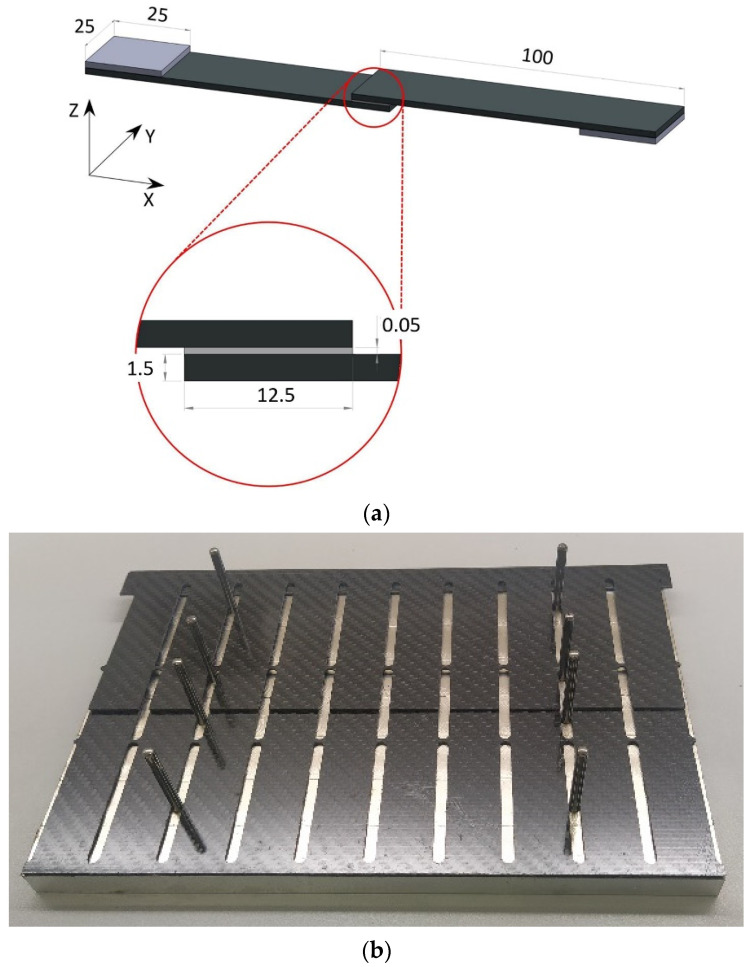
(**a**) Single sample dimensions in mm, and (**b**) bonding jig.

**Figure 3 polymers-15-00121-f003:**
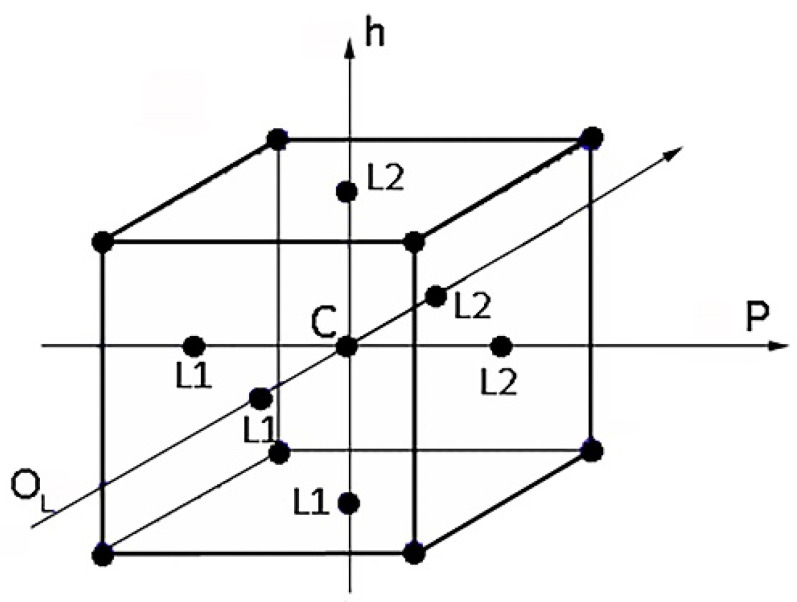
A three-factor layout for face-centered central composite design (FCCD).

**Figure 4 polymers-15-00121-f004:**
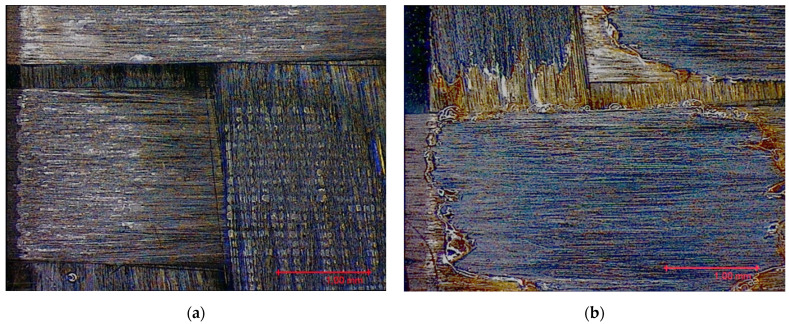
Surface obtained using (**a**) F = 0.15 J/cm^2^ and (**b**) F = 7.95 J/cm^2^.

**Figure 5 polymers-15-00121-f005:**
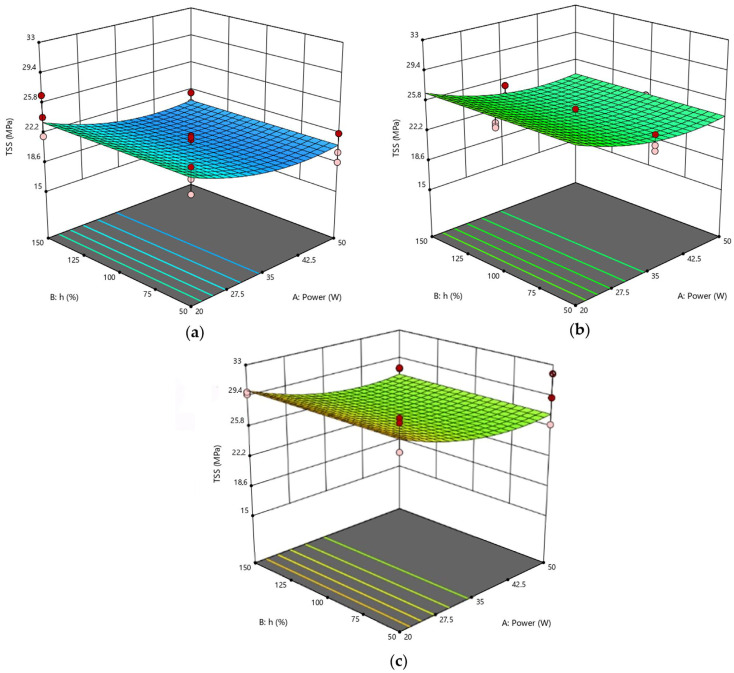
RSD 1: (**a**) O_L_ = 50%; (**b**) O_L_ = 100%; (**c**) O_L_ = 150%.

**Figure 6 polymers-15-00121-f006:**
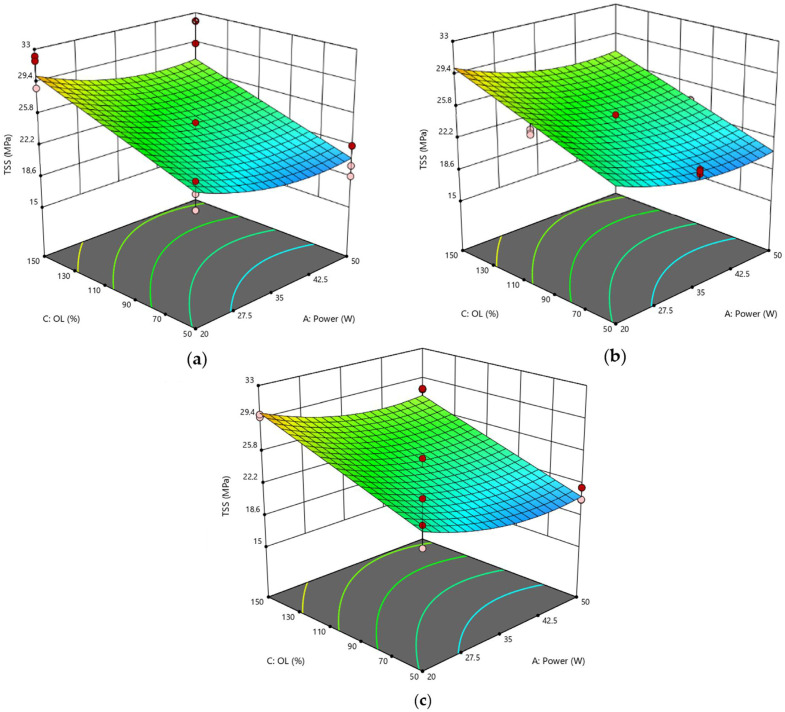
RSD 1: (**a**) h = 50%; (**b**) h = 100%; (**c**) h = 150%.

**Figure 7 polymers-15-00121-f007:**
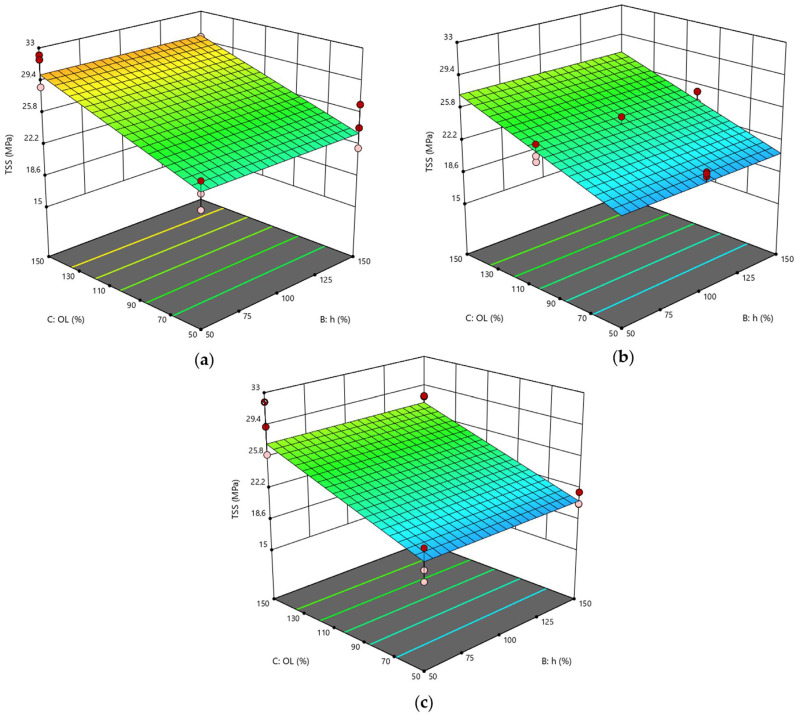
RSD 1: (**a**) P = 20 W; (**b**) P = 35 W; (**c**) P = 50 W.

**Figure 8 polymers-15-00121-f008:**
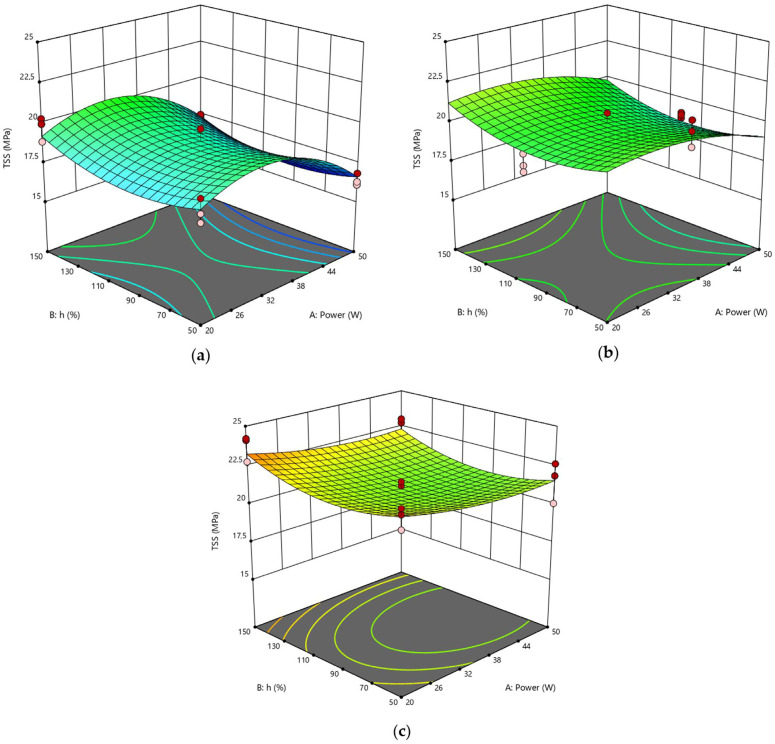
RSD 2: (**a**) O_L_ = 50%; (**b**) O_L_ = 100%; (**c**) O_L_ = 150%.

**Figure 9 polymers-15-00121-f009:**
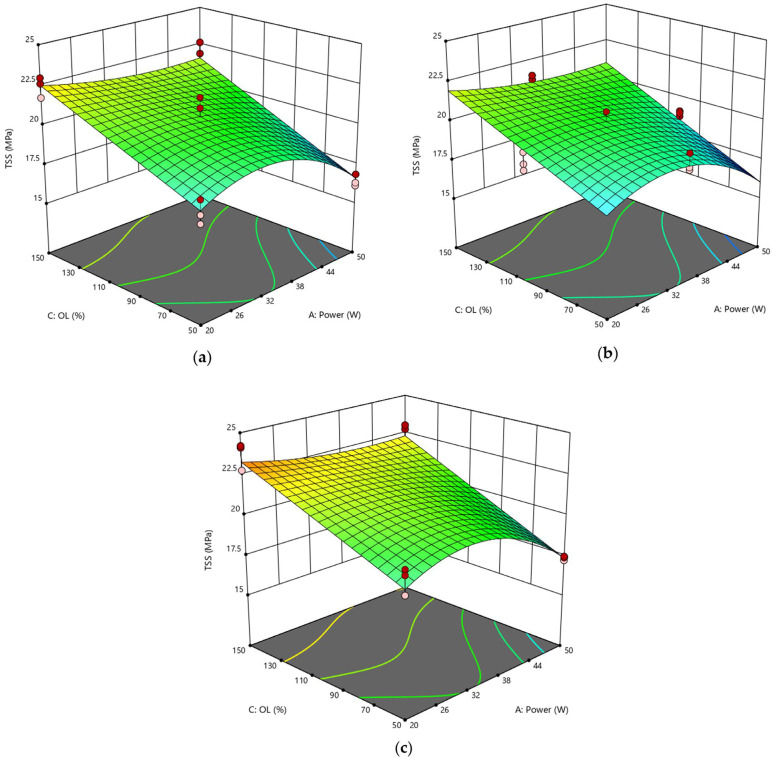
RSD 2: (**a**) h = 50%; (**b**) h = 100%; (**c**) h = 150%.

**Figure 10 polymers-15-00121-f010:**
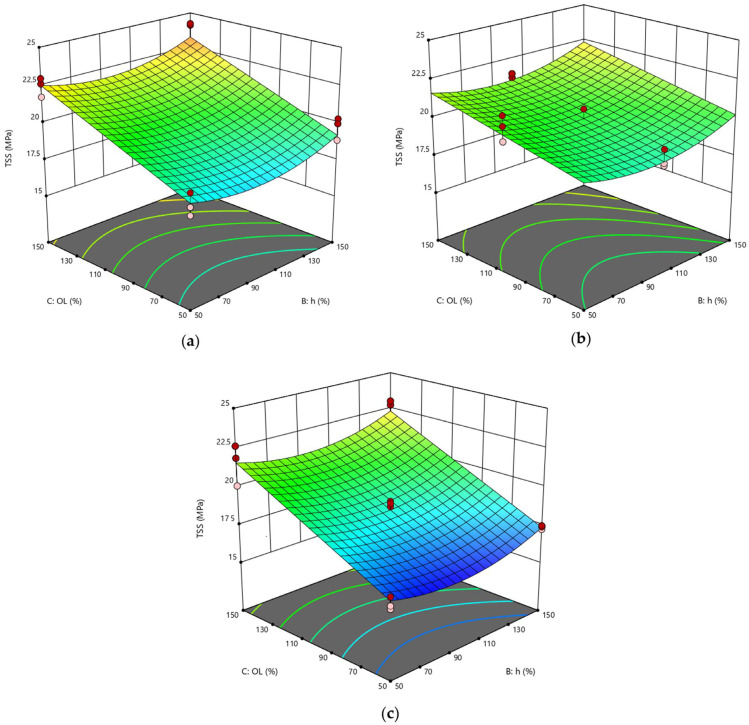
RSD 2: (**a**) P = 20 W; (**b**) P = 35 W; (**c**) P = 50 W.

**Table 1 polymers-15-00121-t001:** Main characteristics and operating range of laser setup.

Characteristic	Symbol	Type/Value	Unit
Operating mode		Pulsed	
Polarization		Random	
Emission wavelength	λ	1064	nm
Nominal maximum power	P_nom_	300	W
Output power tuning		10–100	%
Pulse energy	E_max_	0.2–2	mJ
Pulse frequency	f	2–4000	kHz
Pulse duration	D	20–500	ns
Beam quality (M2)		1.2–1.8	

**Table 2 polymers-15-00121-t002:** Process parameters adopted and levels for laser setup.

	Code	Power, P (W)	Pitch, h (%d)	Lateral Overlap, O_L_ (%d)
Level 1	L1	20	50	50
Midpoint	M	35	100	100
Level 2	L2	50	150	150

**Table 3 polymers-15-00121-t003:** Definition of the experimental designs using focal distance as scenario variable.

	RSD 1	RSD 2
Focus (mm)	−4	0
Focal distance (mm)	184	188
Spot diameter (mm)	0.203	0.05

**Table 4 polymers-15-00121-t004:** Process set-ups experimented with for both scenarios.

	Factor 1	Factor 2	Factor 3
Sample Number	Power, P (W)	Pitch, h (%d)	Lateral Overlap, O_L_ (%d)
1	20	50	150
2	50	50	150
3	50	150	150
4	20	150	150
5	20	150	50
6	50	150	50
7	50	50	50
8	20	50	50
9	20	100	100
10	35	100	50
11	50	100	100
12	35	100	150
13	35	150	100
14	35	50	100
15	35	100	100

**Table 5 polymers-15-00121-t005:** Fluence values corresponding to the three levels of the experimental analysis.

Sample Number	Fluence, F (J/cm^2^)	
	RSD 1	RSD 2
1, 4, 5, 8, 9	0.15	3.18
10, 12, 13, 14, 15	0.27	5.56
2, 3, 6, 7, 11	0.38	7.95

**Table 6 polymers-15-00121-t006:** Morphology of six macro groups sorted according to fluence value.

	Fluence 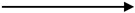
Fluence 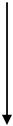		**RSD 1**	**RSD 2**
Sample #1	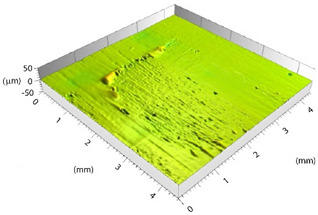	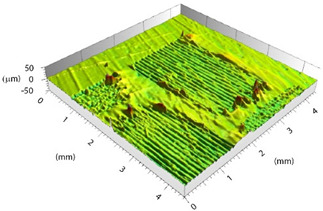
Sample #15	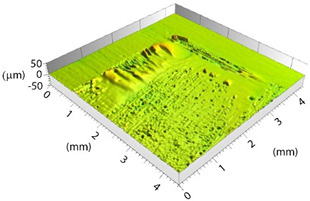	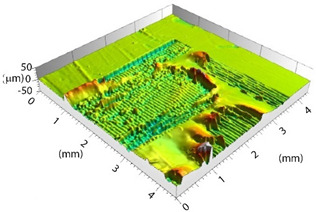
Sample #6	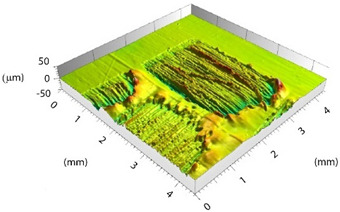	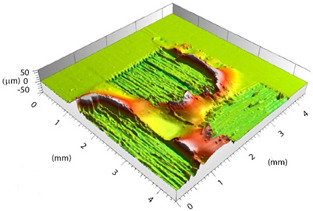

**Table 7 polymers-15-00121-t007:** TSS of the control joints.

Treatment	TSS (MPa)	STD (%)
Degreasing	23.16	28
Abrasion	25.17	5

**Table 8 polymers-15-00121-t008:** Average response in terms of TSS of laser-treated joints for both scenarios.

Sample Number	RSD 1	RSD 2
	TSS (MPa)	TSS (MPa)
1	30.88	22.45
2	29.11	21.53
3	28.08	22.15
4	29.79	23.71
5	24.26	19.71
6	20.89	17.35
7	20.25	16.41
8	23.23	18.32
9	25.87	19.15
10	21.71	19.04
11	23.49	18.91
12	26.72	21.19
13	24.11	20.52
14	23.82	20.99
15	24.06	20.02

**Table 9 polymers-15-00121-t009:** ANOVA for RSD 1.

Source	Sum of Squares	df	Mean Square	F-Value	*p*-Value	
**Model**	390.18	3	130.06	91.46	<0.0001	significant
P	54.89	1	54.89	38.60	<0.0001	
O_L_	309.25	1	309.25	217.46	<0.0001	
P^2^	17.93	1	17.93	12.61	0.0010	
**Residual**	56.88	40	1.42			
Lack of Fit	14.29	11	1.30	0.8843	0.5652	not significant
Pure Error	42.60	29	1.47			
**Cor Total**	447.06	43				

**Table 10 polymers-15-00121-t010:** Significant control factors for RSD 1.

Source	Sum of Squares
O_L_	309.25
P	54.89
P^2^	17.93

**Table 11 polymers-15-00121-t011:** Experimental vs. predicted mean of samples treated with P = 20 W, O_L_ = 150%, and variable h.

	Factor 1	Factor 2	Factor 3	R1	R2	R3	Mean Resp.	Predicted Mean
Sample Number	Power, P (W)	Pitch, h (%d)	Lateral Overlap, O_L_ (%d)	TSS (MPa)	TSS (MPa)	TSS (MPa)	TSS (MPa)	TSS (MPa)
1	20	50	150	31.72	28.70	32.23	30.88	30.1
4	20	150	150	29.87	29.90	29.60	29.79	30.1

**Table 12 polymers-15-00121-t012:** ANOVA for RSD 2.

Source	Sum of Squares	df	Mean Square	F-Value	*p*-Value	
**Model**	158.25	7	22.61	38.49	<0.0001	significant
P	14.62	1	14.62	24.89	<0.0001	
h	4.20	1	4.20	7.14	0.0111	
O_L_	6.97	1	6.97	11.86	0.0014	
Ph	1.21	1	1.21	2.05	0.1603	
P^2^	4.95	1	4.95	8.42	0.0062	
h^2^	7.70	1	7.70	13.11	0.0009	
PO_L_	6.66	1	6.66	11.34	0.0018	
**Residual**	21.74	37	0.5874			
Lack of Fit	6.40	7	0.9137	1.79	0.1268	not significant
Pure Error	15.34	30	0.5113			
**Cor Total**	179.99	44				

**Table 13 polymers-15-00121-t013:** Significant control factors for the RSD 2.

Source	Sum of Squares
P	14.62
h^2^	7.70
O_L_	6.97
P^2^O_L_	6.66
P^2^	4.95
h	4.20
PO_L_	1.21

**Table 14 polymers-15-00121-t014:** Prediction accuracy of the two Response Surface Designs measured through the Tolerance interval values.

	Predicted Mean (MPa)	MSE (MPa^2^)	Tolerance Interval (MPa)	Tolerance Interval (%)
RSD1	30.1	1.420	±3.57	11.9
RSD2	23.3	0.587	±2.28	9.8

**Table 15 polymers-15-00121-t015:** Comparison of TSS values (MP Predicted Mean—CJ).

Scenario	MP Predicted Mean (MPa)	TSS (%) vs. Degreased CJ	TSS (%) vs. Abraded CJ
RSD 1	30.1	29.9	19.5
RSD 2	23.3	0.4	−7.6

## Data Availability

The data presented in this study are available on request from the corresponding author.

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
