# Peer review of "A Response Surface Methodology Approach to Improve Adhesive Bonding of Pulsed Laser Treated CFRP Composites"

_polymers, 2022, doi:10.3390/polym15010121_

Round 1

Reviewer 1 Report

This paper contains extensive study and data regarding A response surface methodology approach to improve adhesive bonding of pulsed laser treated CFRP composites. The paper very welly written and have values insight that can be useful for the scientific community and industry. The methodology is scientific, and the paper is organized and written up to journal standard excluding some minor grammatical error in sentence structure. The article not have enough recent literature survey. I suggest minor revision before considering publication.

Author Response

“This paper contains extensive study and data regarding A response surface methodology approach to improve adhesive bonding of pulsed laser treated CFRP composites. The paper very welly written and have values insight that can be useful for the scientific community and industry. The methodology is scientific, and the paper is organized and written up to journal standard excluding some minor grammatical error in sentence structure. The article not have enough recent literature survey. I suggest minor revision before considering publication.”

Based on Reviewer #1’s comments, the new version of the article has been revised by a native speaker and all typos or grammatical errors were corrected. In addition, more recent literature references on the topic have been included in the text.

Reviewer 2 Report

General comments:

In this submitted paper, response surface methodology was applied to establish the correlation between laser treatment parameters and adhesive bonding strength. This paper exhibited to a clear logical structure and description about experimental result. Although there were limited discussions on improving mechanism, the optimal setting of laser treatment parameters was also important for engineering application, and was well studied in this paper. However, there are still some issues that need to be addressed before publication. It is recommended that this manuscript can be accepted for publication after significant improvement according to comments.

The detailed comments are listed below:

(1)   There were two focal distances of laser used in this study, and there is a big difference in spot diameter, please provide more explanations about it.

(2)   When the focal was -4mm, the laser focus was not on the surface or CFRP material, and there is a question that pulse energy would be completely absorbed by CFRP. It is suggested to measure the received energy on CFRP surface by energy or power meter.

(3)   Table 6 presents the morphologies of CFRP surface from six macro groups, but treated by different energy fluence. When the same laser fluence was applied, would the same morphology be observed for CFRP surface in different groups, e.g. Sample 1# and 8#?

(4)   The key value of significance level α was chosen as 0.05 for RSD 1, please provide more clarification.

(5)   It is difficult to evaluate the prediction accuracy by comparing figures 5-9, and a table should be better to help present this result.

(6)   Tables 9 and 10 present three terms producing a significant effect on the modification of TSS values, namely P, OL, P2 , and the descriptions of the three terms were missed, P for laser power? OL for lateral overlap? How many laser processing parameters were included in the RSD methods?

(7)   The authors stated that “the difference between the two maxima is not surprising because it depends on the fact that the parameter h is not significant in this model.”, and it should be better to compare experimental results between two groups, if they showed the same TSS value, then this statement could be accepted, otherwise the authors should work on the optimization of RSD predication model.

(8)   In Conclusions, the authors stated that “an analysis based on surface and morphological investigation concluded that the fluence parameter should be considered since it also influences heating of the matrix.” Laser fluence was changed by pulse energy, focal and spot diameter, but pitch and lateral overlap also affected heating of CFRP matrix. Moreover, joints treated with the same fluence showed different adhesive bonding strengths, and it indicates that other parameters also exhibited influence on adhesive bonding strength. It is suggested to use average energy density rather than fluence, and please refer to this publication (Journal of Manufacturing Processes 62 (2021) 555565, https://doi.org/10.1016/j.jmapro.2020.12.055).

Author Response

(1)   There were two focal distances of laser used in this study, and there is a big difference in spot diameter, please provide more explanations about it.

and

(2)   When the focal was -4mm, the laser focus was not on the surface or CFRP material, and there is a question that pulse energy would be completely absorbed by CFRP. It is suggested to measure the received energy on CFRP surface by energy or power meter.

and

(3)   Table 6 presents the morphologies of CFRP surface from six macro groups, but treated by different energy fluence. When the same laser fluence was applied, would the same morphology be observed for CFRP surface in different groups, e.g. Sample 1# and 8#?

and

(8)   In Conclusions, the authors stated that “an analysis based on surface and morphological investigation concluded that the fluence parameter should be considered since it also influences heating of the matrix.” Laser fluence was changed by pulse energy, focal and spot diameter, but pitch and lateral overlap also affected heating of CFRP matrix. Moreover, joints treated with the same fluence showed different adhesive bonding strengths, and it indicates that other parameters also exhibited influence on adhesive bonding strength. It is suggested to use average energy density rather than fluence, and please refer to this publication (Journal of Manufacturing Processes 62 (2021) 555–565, https://doi.org/10.1016/j.jmapro.2020.12.055).

R1,2,3,8: We sincerely thank Reviewer #2 for her/his comments as they gave us the opportunity for being more precise in describing and justify the rationale behind the choice of the statistical analysis factors. We chose to analyze two designs with different focal distances because the pulse energy (used to calculate the fluence) is an extremely influential parameter that, if considered a factor rather than a scenario variable, would have hidden the effects of the other factors. However, this is a machine setting on which, based on the device used, it is not possible to act except by setting discrete values; therefore, the only way to operate on the amount of energy contributing to the substrate is to change the spot diameter, thus acting on the focal distance.

We have now reported the following explanation in section 3.3: “In good agreement with the morphological observations reported in Section 3.1, the mechanical behavior observed in the two scenarios reflected the fact that the excessive heating of the material, which caused local combustion and degradation of the resin that negatively affected the surface state, weakened the adhesion interactions at the substrate-adhesive interface and, consequently, lowered the shear strength of joints. However, it should be noted that different values of TSS were obtained from samples belonging to the same macrogroup, suggesting that analysis of the fluence parameter alone cannot provide complete information, as it does not take into account the influence of overlapping laser beams in the longitudinal and transverse directions [Journal of Manufacturing Processes 62 (2021) 555–565, https://doi.org/10.1016/j.jmapro.2020.12.055]. Therefore, further thorough investigation was conducted via statistical method to determine the effect of process parameters such as pitch (h), lateral overlap (OL) and power (P) on the mechanical response (TSS).

(4)   The key value of significance level α was chosen as 0.05 for RSD 1, please provide more clarification.

R4:   The choice of using α = 0.05 as significance threshold for Fisher's tests is dictated by the fact that this is the value suggested by default by the Design Expert tool, as well as the most widely recommended value in the literature. In the specific case studied, moving the threshold from 0.05 to even more precautionary values (e.g., 0.01 for Fisher's first test of the validity of the regression approach or 0.1 for the Lack of Fit test) would not have led to different considerations than those indicated, since the significance of the regression model and factors is confirmed by whatever significance threshold chosen. This can be verified by analyzing the p-values reported in Table 9, which are all <0.0001, i.e., lower than any α value chosen. For the Lack of Fit test, however, the p-value is greater than 0.1, which is the most conservative threshold used to conduct it.

(5)   It is difficult to evaluate the prediction accuracy by comparing figures 5-9, and a table should be better to help present this result.

R5: It is common practice to use the 95% prediction interval to quantify the uncertainty involved and evaluate the prediction accuracy of the regression model and to interpret such an interval as if 95% of the future responses will be contained within this interval. However, this is inappropriate, and it can result in a gross underestimation of the real uncertainty of the predicted response. To correctly quantify the uncertainty of a prediction of a regression model, so-called tolerance intervals should be used. The tolerance interval can be calculated using the Mean Square Error (MSE) value resulting in the ANOVA tables (Tables 9 e 12, respectively for RSD1 e RSD2). The MSE of RSD1 is 1.420, while the MSE of RSD2 is 0.587, from which the limits of the relative tolerance intervals can be calculated as . It results that the model accuracy of RSD2 is higher than that of RSD2. As suggested by Reviewer #2, an explanatory table (new Table 14) has been included in Section 3.3.2.

(6)   Tables 9 and 10 present three terms producing a significant effect on the modification of TSS values, namely P, OL, P2, and the descriptions of the three terms were missed, P for laser power? OL for lateral overlap? How many laser processing parameters were included in the RSD methods?

R6:   Although the meanings of P, OL, and h parameters had been already defined on page 6, Section 2.3, we have also reported them in the Nomenclature section.

(7)   The authors stated that “the difference between the two maxima is not surprising because it depends on the fact that the parameter h is not significant in this model.”, and it should be better to compare experimental results between two groups, if they showed the same TSS value, then this statement could be accepted, otherwise the authors should work on the optimization of RSD predication model.

R7:   Since the process factor h resulted not significant in the ANOVA table (see Table 10), this means that in the range of variation considered (from 50 to 150%d) it does not influence the TSS value.

Factor 1

Factor 2

Factor 3

R1

R2

R3

Mean Resp.

Predicted Mean

Sample number

Power, P (W)

Pitch, h (%d)

Lateral overlap, OL (%d)

TSS (MPa)

TSS (MPa)

TSS (MPa)

TSS (MPa)

TSS (MPa)

1

20

50

150

31.72

28.70

32.23

30.88

30.1

4

20

150

150

29.87

29.90

29.60

29.79

30.1

For this reason, the predicted TSS value – i.e., the value from the regression model obtained by varying h from the low level to the high level – is the same at parity of values assumed by the other two factors. Since the optimal condition is given by P = 20 W and OL = 150%, regardless of h, the mean value of the TSS does not vary and therefore the same predicted mean is obtained. Although the experimental values are slightly different from each other, from a statistical point of view, the ANOVA conducted actually shows that the recorded point variations are only due to pure experimental error (i.e., background noise) and not attributable to the variation in h. As suggested by Reviewer #2, the above explanatory table has been included in Section 3.3.1 (Table 11). 

Reviewer 3 Report

Dear Authors,

Here are a few comments for improvement,

1.)  Introduction – the need/novelty of the research is not well laid out. Although L54-73 reflect the importance of research to some extent, however, requires restructuring.

2.)  L75 -117 – split into two paragraphs and shorten each. Be more precise and concise.

3.)  L120 – 125 -please state how you have chosen the parameters to manufacture the composites. Optimised or from a previous study? Justify.

4.)  L163 – be specific on how the image processing has been carried out.

5.)  L170 –  how the parameters were selected for mechanical characterisation?

6.)  L213 – rewrite to – Analysis of Variance

7.)  L485 -496 - sounds like the introduction and setting up the importance of research. Therefore, considering restructuring the introduction is recommended. 

Many thanks,

Author Response

(1)  Introduction – the need/novelty of the research is not well laid out. Although L54-73 reflect the importance of research to some extent, however, requires restructuring.

and

(2)  L75 -117 – split into two paragraphs and shorten each. Be more precise and concise.

and

(7)  L485 -496 - sounds like the introduction and setting up the importance of research. Therefore, considering restructuring the introduction is recommended. 

R1,2,7: the introductory section has been restructured according to the Reviewer’s recommendations and divided into two separate paragraphs. Specifically, paragraph 1.2 now reports some considerations previously present in section 3.4.

(3)  L120 – 125 -please state how you have chosen the parameters to manufacture the composites. Optimised or from a previous study? Justify.

R3: Based on the Reviewer’s comment, on page 4 it is now reported: “Based on the results of preliminary tests aimed at optimizing the polymerization process, CFRP panels of thickness of 1.50 ± 0.02 mm and Young’s modulus E equal to 70 ± 5 GPa were obtained using a vacuum bag in an autoclave for 2 h at 135°C and pressure of 6 bar.

(4)  L163 – be specific on how the image processing has been carried out.

R4: Image processing via X-Pro software consisted in the digitization of images acquired under the microscope, addition of appropriate markers in relation to the magnification of the lens used, and any adjustments (such as contrast and brightness) to improve image quality. A mention of this has been added to the main text in Section 2.2.

(5)  L170 –  how the parameters were selected for mechanical characterisation?

R5: The reference standard followed for mechanical testing of joints is now mentioned on page 5, Section 2.2.

(6)  L213 – rewrite to – Analysis of Variance

R6: The suggested change was met.

Round 2

Reviewer 2 Report

The authors have properly revised their manuscript.

Reviewer 3 Report

The changes has been and now can be accepted